# Exploring the Mechanisms Underlying Petal Pigmentation Differences in Two Cultivars of *Physalis philadelphica* Based on HPLC and NGS

Hongyu Qiao [1,†], Wennan Zhao [1,†], Song Tian [2], Da Wang [1], Haiyan Wu [1], Chenyu Wang [1], Jiaming Zhu [1], Nan Li [3], Xu Zhu [3], Shujing Mu [3], Jingying Zhang [1,*] and Hongxia He [3,*]

1   Modern Vegetable Industry Technology and Germplasm Resource Innovation Team, Northeast Asia Special Germplasm Resource Conservation and Innovation Center Vegetable Breeding Technology Innovation Team, College of Horticulture, Jilin Agricultural University, Changchun 130118, China; qiaohongyu@jlau.edu.cn (H.Q.); 20231415@mails.jlau.edu.cn (W.Z.); 2114100408@mails.jlau.edu.cn (D.W.); wuhaiyan@jlau.edu.cn (H.W.); 20231418@mails.jlau.edu.cn (C.W.)
2   Jilin Provincial Research Institute of Vegetables and Flowers, Changchun 130119, China; tiansongaabbcc@163.com
3   Institute of Agricultural Biotechnology, Jilin Academy of Agricultural Sciences (Northeast Agricultural Research Center of China), Jilin Provincial Key Laboratory of Agricultural Biotechnology, Changchun 130033, China; tinbon27@163.com (N.L.); zhuxu0615@163.com (X.Z.); msj900605@163.com (S.M.)
*   Correspondence: zhangjingying@jlau.edu.cn (J.Z.); hongxia_365@163.com (H.H.)
†   These authors contributed equally to this work and are co-first authors.

**Abstract:** *Physalis philadelphica*, a member of the Solanaceae family, commonly known as Physalis, is a one-year-old herbaceous plant with both medicinal and edible properties, as well as ornamental value. At present, only limited research is available on the flower color of *P. philadelphica*. This study aimed to elucidate the metabolic characteristics underlying the flower color of *P. philadelphica* and to identify key genes associated with flower color metabolism. We selected two representative varieties of *P. philadelphica* with significant differences in flower color, namely, "Tieba" (yellow flower) and "Qingjin" (yellow-purple flower), as the experimental materials. The analysis of related pigment components and the determination of relative content by high-performance liquid chromatography were conducted to investigate the flower color-related metabolic pathways of *P. philadelphica*. Through next-generation sequencing, these pathways were further investigated for the characteristics and differentially expressed genes (DEGs) associated with flower color formation. The results of the research show that: Anthocyanin is the main component of petal coloring of *P. philadelphica* var. Qingjin, while malvidin pigment, pelargonidin, delphinidin, and cyanidin are the main components of flower color intensity. Carotenoids are the main components of the petal coloring of *P. philadelphica* var. Tieba and β-carotene is the main component of flower color intensity. Comparing different developmental stages of these two kinds of Physalis pubescens, we identified two key transcription factors (TFs) (eBP and STAT) that were involved in the inhibition of anthocyanin synthesis and regulate the inhibition of pf05G124640 (dihydroflavonol 4-reductase) and pf09G224140 (anthocyanin synthase) in anthocyanin synthesis. One heat shock transcription factor was found to regulate the flavonoid and flavonol synthesis pathway of pf01G020090 (anthocyanin 3-O-glucosyltransferase); two key TFs (NAC and G2-Like), pf10G255070 (isoricin dehydrogenase) and pf09G237080 (abscisic acid 8'-hydroxylase), played important roles in carotene biosynthesis. This study provides new insights for further exploration of the genetic diversity of petal coloring in *P. philadelphica* and establishes a foundation for subsequent molecular breeding efforts.

**Keywords:** *Physalis philadelphica*; petal coloring; anthocyanin; carotenoid; transcription factor





## 1. Introduction

*P. philadelphica*, also known as "yellow tomatillo" and "foreign tomatillo", belongs to the *Solanum* genus of herbaceous plants in the family of Solanaceae. *P. philadelphica* is a native fruit of Mexico and Central America [1], which possesses high edible, medicinal, and ornamental values [1,2] and is now cultivated in Jilin and Heilongjiang provinces of China [3]. The predominant color of its flowers, typically yellow or yellowish, serves as a significant indicator of both its ornamental and commercial value. Previous studies have shown that flower color is affected by a variety of factors; apart from genetic and environmental variables, the pH levels within the vesicle, plant hormones such as gibberellins, the intracellular environment of the plant, and the internal or superficial structure of the petals also play significant roles in determining flower color. However, all these factors ultimately exert their influence on pigments, thereby influencing flower color. Therefore, the decisive factor in determining plant flower color remains the type and concentration of the pigments present [4,5].

The diversity of flower colors primarily arises from the activities of three pigments, namely flavonoids, carotenoids, and betalains. Betalains are unique to Caryophyllales, whereas flavonoids and carotenoids are widespread throughout the plant kingdom [6–9]. Flavonoids encompass various subgroups including flavones, flavonols, flavanones, and anthocyanins [10–13]. Among these, anthocyanins are the most abundant polyphenols and are chiefly responsible for petal coloration, primarily under genetic regulation. Carotenoids, another essential pigment group, are abundant in the petals of higher plants and significantly influence petal color. Typically found as lipids within petals, their biosynthesis pathways involve geranylgeranyl diphosphate synthesis, octahydrotomato lycopene synthesis, and octahydrotomato lycopene desaturation. Extensive research has focused on these pathways, elucidating the corresponding enzymes and genes involved.

The present study initially employs high-performance liquid chromatography (HPLC) for the analysis of pigment components and the determination of their relative contents. Furthermore, transcriptomic analysis was conducted to elucidate the metabolic pathways associated with flower color, aiming to clarify the metabolic characteristics of flower color for *P. philadelphica* and obtain crucial genetic information regarding color metabolism. A bioinformatics analysis system for *P. philadelphica* flower color was established, laying the groundwork for subsequent molecular breeding efforts.

## 2. Materials and Methods

### 2.1. Materials and Experimental Design

The test materials were *P. philadelphica* var. Tieba (A; yellow flower) and *P. philadelphica* var. Qingjin (B; yellow and purple flower). All the plants used in this study were grown in the Vegetable Research and Teaching Base of Jilin Agricultural University. Each experimental area consisted of seven trees planted with a row spacing of 120 cm and a plant spacing of 50 cm. The experimental design followed a randomized area group design, with each experiment replicated three times. The two-stage selection (S2: Initial flowering stage; S3: Full flowering stage) is shown in Figure 1 with three biological replicates. A total of 12 samples were frozen in liquid nitrogen and transported to Lianchuan Biological Company for transcriptome database construction and analysis.

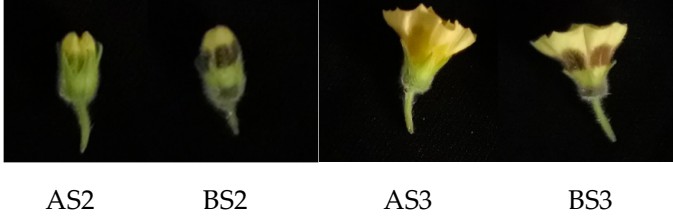

|  AS2  |  BS2  |  AS3  |  BS3  |

**Figure 1.** Coloring of petals of two kinds of *P. philadelphica*. A: Tieba; B: Qingjin; S2: Initial flowering stage; S3: Full flowering stage.

*2.2. Determination of Pigment Content and Transcriptome Sequencing*

2.2.1. Determination of Total Anthocyanin Content

The absorbance readings of the extracted solution were measured at wavelengths of 520 nm, 620 nm, and 650 nm, following the protocols outlined in the manual for determining total plant anthocyanin content. The calculation formula is as follows:

$$\text{Total anthocyanin content: } Q = A\lambda \times V \times 1000/489.72 \, M \, (\text{mmol/g FW}) \tag{1}$$

where $A\lambda = (A530 - A620) - 0.1 (A650 - A320)$; V represents the volume of the extract and M represents the weight of the fresh sample; 95% ethanol (with 0.1 mol/L HCl) was used as a blank control.

2.2.2. Determination of Carotenoid Content

Carotenoid content was assessed following the specified procedures for carotenoid determination, with absorbance measured at 440 nm using a spectrophotometer. The calculation formula is as follows:

$$\text{Carotenoid content: } (\text{mg/g}) = A440 \times V \text{ mention} \times 1000 \times D/\varepsilon \times d \times W = 0.04 \times A440 \times D/W \tag{2}$$

where V mention represents the total volume of the sample after the extract (10 mL); $\varepsilon$ represents the carotenoids experience extinction coefficient (250 L/g/cm); d represents the cuvette aperture (1 cm); D represents the dilution times; W represents the sample mass (g); 1000 represents the unit conversion factor (1 g = 1000 mg).

2.2.3. Determination of Anthocyanin and β-Carotene Contents

The contents of anthocyanin and β-carotene were determined by using RigolL3000 HPLC with Sepax Bio-C18 reversed-phase column (250 mm × 4.6 mm, 5 μm) in accordance with the instructions of the phycocyanin and β-carotene kits, respectively. The metabolite data were analyzed using DPS7.5 software.

2.2.4. Transcriptome Sequencing

Total RNA was extracted from the petals of the two varieties of *P. philadelphica* using a plant total RNA extraction kit (Kumei Biotechnology Co., Ltd., Changchun, Jilin, China). RNA integrity and potential contamination were assessed through agarose gel electrophoresis. Subsequently, RNA purity (OD260/OD280 ratio) was determined by Nanodrop, and the RNA concentration was accurately quantified by Qubit. The final RNA integrity was evaluated by Agilent 2100 [14–16]. Following quality control measures, HiSeq sequencing was performed, and the preprocessed data were filtered to obtain Cleanreads [17]. Subsequently, alignment with the reference genome was performed using Hisat2 [18], while gene location information specified in the genome annotation file (gtf, gene transfer format, a commonly used format for annotating genes on chromosomes) was separately analyzed to delineate gene positions on the chromosomes. The transcripts were reconstructed and the expression levels of all genes in each sample were calculated using Stringtie [19,20]. The genome assembly used for transcript quantification is derived from https://ngdc.cncb.ac.cn/gwh/Assembly/9765/show (accessed on 22 June 2022).

Gene expression in the *P. philadelphica* samples was analyzed following the FPKM standardization, and differentially expressed genes (DEGs) were screened based on both the multiplicity of difference and the significant level [21]; The criteria were: fold difference $FC \geq 2$ or $FC \leq 0.5$ (i.e., the absolute value of $\log_2 FC \geq 1$) and q value < 0.05. Identification of differentially expressed genes using DESeq2_1.22.2 software. Genes were clustered and analyzed using a differential gene clustering heatmap (Heatmap) to reflect clustering expression patterns more intuitively among the three replicates of each sample, employing Z scores ($Z_{\text{sample-i}} = [(\text{FPKM}_{\text{sample-i}}) - \text{Mean}_{(\text{FPKM of all samples})}]/[\text{Standard deviation}_{(\text{FPKM of all samples})}]$) for gene expression. The heatmap displays samples on the *x*-axis and the top 100 differentially expressed genes with the smallest Q-values

on the *y*-axis. The color gradient from blue to white to red represents low to high gene expression levels, allowing for visual comparison of the expression levels of the same gene across different samples. The set of DEGs was subsequently analyzed by GOseq_1.34.1 and KOBAS 3.0 software for GO functional enrichment and KEGG pathway enrichment, respectively. Venn diagrams, volcano plots, and heatmaps were generated using the Lianchuan Bio Cloud Platform (https://www.lc-bio.cn/overview, (accessed on 29 June 2022)).

2.2.5. Single Nucleotide Polymorphism (SNP) and Indel Analyses

SNP sites in coding regions were analyzed based on the transcriptome level. Based on the results of the Hisat2 comparison between each sample and the reference genome, the potential SNP and indel information of each sample was obtained by the mpileup process using the SAMtools 1.14 software and then annotated with ANNOVAR [22–24].

2.2.6. Validation of Next-Generation Sequencing (NGS) by Real-Time Quantitative PCR (qRT-PCR)

The plant total RNA extraction was performed according to the instructions of the RNAprep Pure kit (Kumei Biotechnology Co., Ltd., Jilin, China). The cDNA was obtained by reverse transcription according to the instructions of the TransScript® One-Step gDNA Removal and cDNA Synthesis SuperMix Reverse Transcription Kit (TaKaRa, Beijing, China). The primers for the internal reference genes were designed by the Primer 6.0 software and commissioned to Jilin Kumei Biological Co. The primer sequences are shown in Table 1. The cDNAs were amplified by using the PerfectStart® Green qPCR SuperMix kit (Transgen Biotch Co., Beijing, China) and the qRT-PCR instrument. The primer specificity was confirmed by the solubility curve. The average Ct value was normalized with respect to the internal reference. The gene expression levels in different *P. philadelphica* varieties were analyzed by using Pp.GAPDH [25,26] as the internal reference gene and calculated by the $2^{-\Delta\Delta Ct}$ method.

**Table 1.** Primer sequences.

| Gene IDs | Forward Primers (5′-3′) | Reverse Primers (5′-3′) |
|---|---|---|
| pf05G124640 | GCTTCTTTGTCTTGTCCGTTGTCTG | TGCCATTGAGACTTGCCGACAG |
| pf09G224140 | GCTTGGCTAGGAGTGGCATAAAGG | TCGTGGGTACTTGAGGTCCTTCATC |
| pf05G137060 | GTGTGGTGGTGGCGGATATGC | CTGTCATCAGTTCCACGACCATGTC |
| pf01G020090 | GAGTTGGATAGGCTTTCGGCTGAG | AATGGCTCCCTCAAATGGCTGATG |
| pf05G120860 | CTTGGAACCTTATGACCACCTCTGC | TTGCCATCATCAGGAAGAGCCATG |
| pf06G177100 | GCCAGTTTGACCACCCTCATTCTC | CAGGGAGGTTCAGCAGGAATAGAAG |
| pf06G169640 | ACATGGTACGGCAAATCATCCTCAG | CTCGTGGTCTCATTGGTCTGGTTG |
| pf09G237080 | ATTGGTGTCATCTTTGCAGCTAGGG | GTGACGGCGAGTAGGACACTAGG |
| pf05G143230 | TGGAGCCAGTGGTCGAAGGTC | ATCCAAGTGCGATGTCCAAGTATCC |
| pf01G016330 | GCTGTTGTGTCCGAGAACGAAGAG | GTCCCCATCAAGTAGTGCAAACCC |
| pf11G267170 | AAGCCAGAATGAGCATGAGCAGAG | AGCCAGTGTCACCATCAGCAATG |
| pf06G179690 | AACGGATTGCCCTCGACTGAAAC | AGCCTCAATACCCTTTGCCAACG |
| *Pp.GAPDH* | TGTGGGTGTCAACGAGAAGGAATAC | ATAAGACCCTCCACAATGCCAAACC |

## 3. Results

*3.1. Analyses of Color-Related Components*

3.1.1. Total Anthocyanin Content

The anthocyanin content of the two varieties of *P. philadelphica* (Tieba and Qingjin) was determined at the initial (S2) and full (S3) flowering stages (Table 2). The anthocyanin content in Qingjin reached the maximum 57.9 nmol/g at the initial flowering stage (BS2) and 50.69 nmol/g at the full flowering stage (BS3), which were both significantly higher than those of Tieba; the difference between the maximum and minimum total anthocyanin content was 52.31 nmol/g. These results demonstrated a significantly lower total antho-

cyanin content in Tieba flowers and suggested that anthocyanin is the primary pigment responsible for the coloration of Qingjin petals.

**Table 2.** Anthocyanin contents (the average of three biological replicates) of two kinds of *P. philadelphica* at two flowering stages.

| Flowering Stages | Anthocyanin Content (nmol/g) |
|---|---|
| Tieba initial flowering stage (AS2) | 5.59 [b] ± 0.48 |
| Qingjin initial flowering stage (BS2) | 57.90 [a] ± 0.32 |
| Tieba full flowering stage (AS3) | 11.72 [b] ± 1.27 |
| Qingjin full flowering stage (BS3) | 50.69 [a] ± 0.09 |

Note: Different letters (a and b) indicate $p < 0.05$.

### 3.1.2. Carotenoid Content

The carotenoid content of the two varieties of *P. philadelphica* was measured at the two flowering stages (Table 3). The highest content of carotenoid in Qingjin was found in the initial flowering stage (BS2), at 2.84 mg/g, which was significantly higher than that in the full flowering stage (BS3), as 2.01 mg/g. For Tieba, 1.45 mg/g and 1.40 mg/g carotenoids were detected in the two flowering stages (AS2 and AS3), respectively. The results revealed a higher carotenoid content at the initial flowering stage compared to the full flowering stage, suggesting that carotenoid undergoes partial degradation and may also participate in other physiological processes as the flower develops. This implied that carotenoid may not be the sole contributor to the final coloration of the petals.

**Table 3.** Carotenoid contents (the average of three biological replicates) of two kinds of *P. philadelphica* at two flowering stages.

| Flowering Stages | Carotenoid Content (mg/g) |
|---|---|
| Tieba initial flowering stage (AS2) | 1.45 [c] ± 0.03 |
| Qingjin initial flowering stage (BS2) | 2.84 [a] ± 0.06 |
| Tieba full flowering stage (AS3) | 1.40 [c] ± 0.10 |
| Qingjin full flowering stage (BS3) | 2.01 [b] ± 0.09 |

Note: Different letters (a, b, and c) indicate $p < 0.05$.

### 3.1.3. Delphinidin, Cyanidin, Pelargonidin and Malvidin Contents

The contents of delphinidin, cyanidin, pelargonidin, and malvidin of the two varieties of *P. philadelphica* at the two flowering stages were determined by HPLC (Table 4). We observed that the contents of these four pigments were below the detection limit at both flowering stages of Tieba, suggesting that these pigments were present in low quantities in their petals. Thus, it is presumed that these pigments may not be the primary contributors to the coloration of Tieba petals. At both flowering stages of Qingjin, delphinidin content was notably higher, with 14.04 μg/g at the initial flowering stage and 13.57 μg/g at the full flowering stage, significantly surpassing the content of other pigments. Following delphinidin, cyanidin content was observed to be 1.48 μg/g at the initial flowering stage and 1.42 μg/g at the full flowering stage. Malvidin content was recorded at 1.20 μg/g during the initial flowering stage and 0.88 μg/g at the full flowering stage of Qingjin, while pelargonidin content remained below the detection limit at both flowering stages of Qingjin. In summary, while the primary cause of Tieba petal coloration is not attributed to these four pigments, the dominance of delphinidin content in Qingjin significantly influences petal color, with other pigment contents playing a lesser role.

**Table 4.** Delphinidin, Cyanidin, Pelargonidin, Malvidin contents (the average of three biological replicates) of two kinds of *P. philadelphica* at two flowering stages.

| Flowering Stages | Delphinidin (μg/g) | Cyanidin (μg/g) | Pelargonidin (μg/g) | Malvidin (μg/g) |
|---|---|---|---|---|
| Tieba initial flowering stage (AS2) | Below detection limit | Below detection limit | Below detection limit | Below detection limit |
| Qingjin initial flowering stage (BS2) | 14.04 [a] ± 0.01 | 1.48 [a] ± 0.04 | Below detection limit | 1.20 [a] ± 0.02 |
| Tieba full flowering stage (AS3) | Below detection limit | Below detection limit | Below detection limit | Below detection limit |
| Qingjin full flowering stage (BS3) | 13.57 [a] ± 0.04 | 1.42 [a] ± 0.03 | Below detection limit | 0.88 [b] ± 0.11 |

Note: Different letters (a and b) indicate $p < 0.05$.

### 3.1.4. β-Carotene Content

The content of β-carotene in the two varieties of *P. philadelphica* was also determined at the two flowering stages (Table 5). The highest content of β-carotene was observed as 110.892 μg/g in Tieba at the initial flowering stage (AS2), followed by 108.0468 μg/g at the full flowering stage of Tieba (AS3). For Qingjin, 83.5198 μg/g and 45.6094 μg/g β-carotene were observed in the initial flowering stage (BS2) and full flowering stage of (BS3) of Qingjin. The results indicated that Qingjin accumulated significantly less β-carotene content than Tieba, suggesting that β-carotene, as a carotenoid pigment, is more likely to influence the coloration of Tieba petals.

**Table 5.** β-carotene content (the average of three biological replicates) of two kinds of *P. philadelphica* at two flowering stages.

| Flowering Stages | β-Carotene Content (μg/g) |
|---|---|
| Tieba initial flowering stage (AS2) | 83.5198 [b] ± 0.02 |
| Tieba full flowering stage (AS3) | 108.0468 [a] ± 0.03 |
| Qingjin initial flowering stage (BS2) | 45.6094 [c] ± 0.03 |
| Qingjin full flowering stage (BS3) | 110.8920 [a] ± 0.02 |

Note: Different letters (a, b, and c) indicate $p < 0.05$.

### 3.2. Transcriptomic Analyses

#### 3.2.1. High-Throughput Sequencing Results

A total of 56.30 Gb clean sequencing data were obtained from the cDNA library construction using petal RNA samples from the two varieties of *P. philadelphica* (Tieba and Qingjin) at the early (S2) and full (S3) flowering stages, with each sample yielding 5.95 Gb clean data. The percentage of Q30 bases exceeded 95.08% in all samples. The efficiency of reads mapping to the reference genome ranged from 86.26% to 88.55%. Comparative analysis with the reference genome enabled the prediction of alternative splicing events, optimization of gene structures, and discovery of new genes. Differential gene analysis was conducted using official annotations, resulting in the identification of 7407 new genes, with 3342 of them being functionally annotated. All data have been uploaded to NCBI.

#### 3.2.2. Identification of DEGs

The analysis of FPKM values from transcriptome data involved using the Venn diagram function in the Lianchuan Cloud tool platform to visualize the intersections and unique DEGs associated with different *P. philadelphica* varieties and petal coloration stages. The Venn diagram illustrated comparisons including AS3 vs. AS2, BS3 vs. BS2, AS3 vs. BS2, and AS2 vs. BS2, resulting in 3304, 4181, 762, and 487 entries in the respective comparative groups (Figure 2).

## Differential Expressed Genes

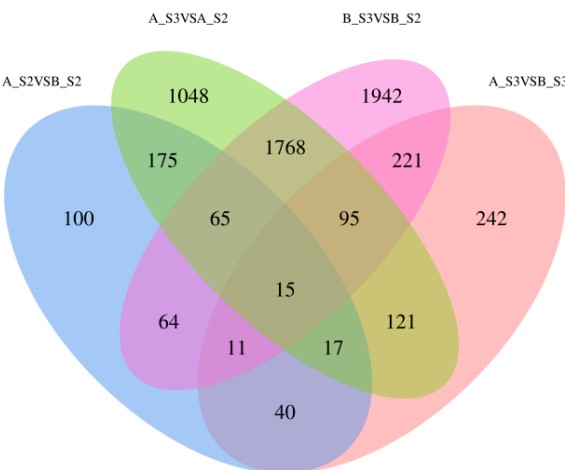

**Figure 2.** Venn diagram showing differentially expressed genes in each comparison group.

The highest number of DEGs were found from S3 vs. S2 comparisons (Figure 3), with 1849 upregulated and 1455 downregulated for AS3 vs. AS2 and 1703 upregulated and 2478 downregulated for BS3 vs. BS2, respectively. In addition, overlapping DEGs were identified among AS3 vs. AS2, BS3 vs. BS2, AS3 vs. BS2, and AS2 vs. BS2 comparison groups, indicating their potential key roles in petal color expression across different varieties (Figure 2). These DEGs were further analyzed for functional annotation and enrichment.

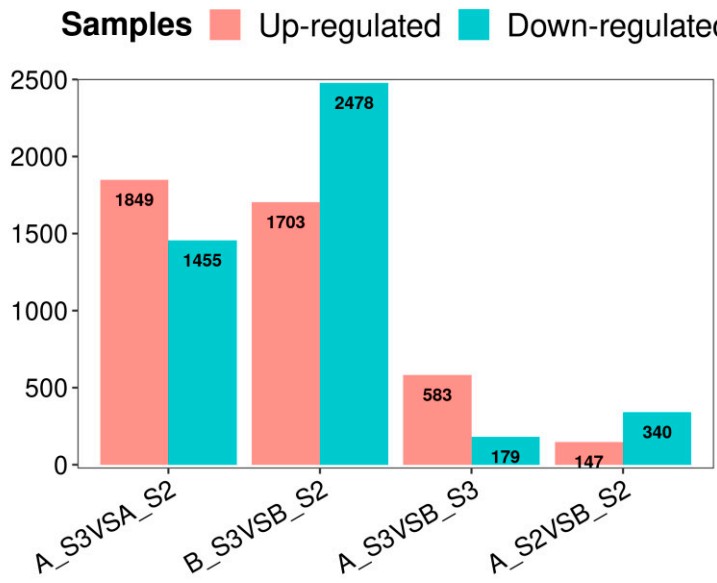

**Figure 3.** Numbers of upregulated and downregulated genes in each comparison group.

The overall distribution of DEGs is shown in Figure 4, with the top 20 genes in AS2 vs. BS2, AS3 vs. BS3, AS3 vs. AS2, and BS3 vs. BS2 comparison groups plotted using the gene_ID, log2(fc), and qval columns in the advanced volcano plot, respectively. Meanwhile, the differential gene clustering heatmap (Figure 5) was used to cluster and analyze the DEGs based on the similarity of gene expression profiles of the samples.

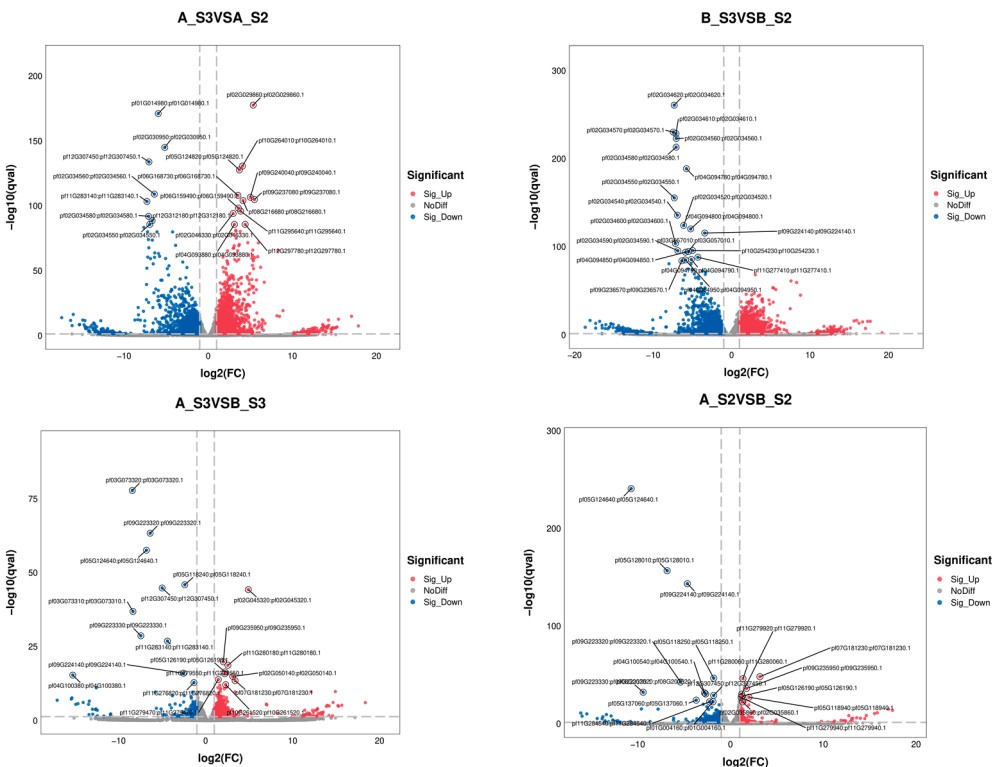

**Figure 4.** Volcanic diagram for differentially expressed genes.

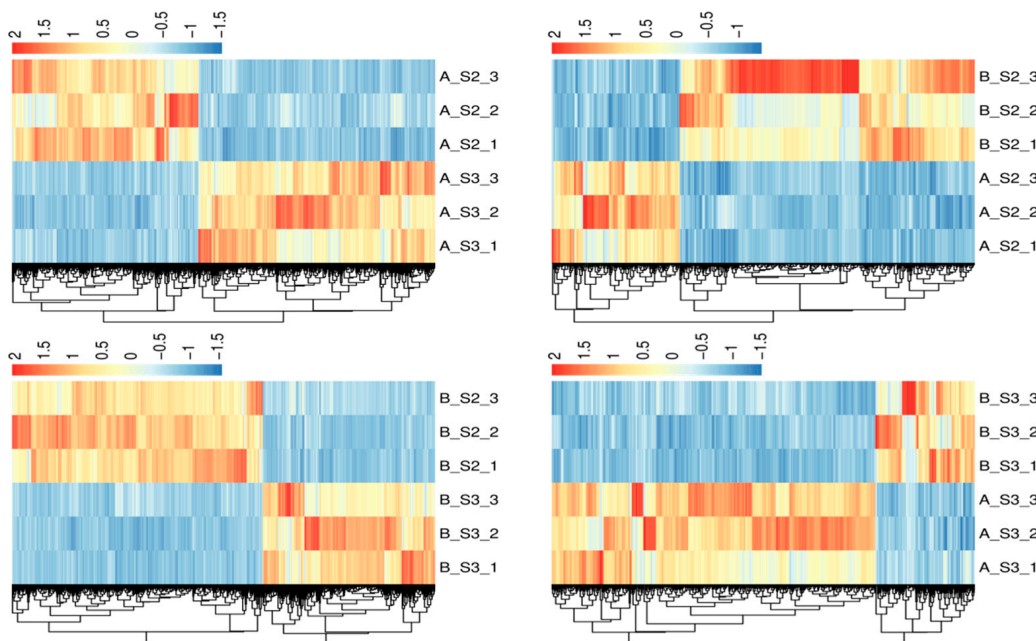

**Figure 5.** Heatmaps of differential genes clustering.

### 3.2.3. GO Enrichment and KEGG Analysis of Petal Coloring-Related DEGs in *P. philadelphica*

The transcriptome data from the two varieties of *P. philadelphica* (Tieba and Qingjin) at initial (S2) and full (S3) flowering stages were screened for DEGs following functional annotation. Based on the GO and KEGG databases, the enrichment analysis was carried out according to the Q-value of the top 20 cellular components, molecular functions, biological processes, and signaling pathways in which the DEGs might be involved (Figures 6 and 7).

The GO analysis showed enrichments of 3304 DEGs in the AS3 vs. AS2 comparison group on 2247 GO terms for molecular functions, cellular components, and biological

processes involved, 4181 DEGs in the BS3 vs. BS2 comparison group in 2652 GO terms, 762 DEGs in the AS3 vs. BS3 comparison group on 1043 GO terms, and 487 DEGs in the AS2 vs. BS2 comparison group on 704 GO terms.

According to the KEGG classification, DEGs were mainly concentrated in flavonoid biosynthesis (map00941), flavonoids and flavonols biosynthesis (map00944), and carotenoid biosynthesis (map00906) (Table 6), which also suggests that these three types of metabolism are associated with complex molecular biological effects on petal coloration.

**Table 6.** Differentially expressed genes related to petal coloration in two varieties of *P. philadelphica.*

| Group | Gene Name | Description | Pathway |
|---|---|---|---|
| *P. philadelphica* Qingjin | pf09G224140 | Anthocyanin synthase | |
| | pf05G124640 | Dihydroflavonol 4-reductase | |
| | pf05G125540 | Flavanone 3-hydroxylase | |
| | pf08G220750 | Flavonol synthase/Flavanone 3-hydroxylase | Flavonoid |
| | pf07G186580 | Chalcone-flavonoid isomerase 3 subtype X1 | |
| | pf09G234590 | Bright blue pigment dioxygenase | |
| | pf08G220770 | Flavonol synthase/Flavanone 3-hydroxylase | |
| | pf01G020090 | Anthocyanin 3-O-glucosyltransferase | Flavonol |
| | pf02G043370 | $\beta$-carotene hydroxylase | |
| | pf03G054950 | Decatriol dehydrogenase | |
| | pf09G235870 | Abscisic acid 8$'$-hydroxylase | |
| | pf09G237080 | Abscisic acid 8$'$-hydroxylase | |
| | pf12G300180 | Decatriol dehydrogenase | |
| | pf10G255070 | Isoprenol dehydrogenase | Carotenoid |
| | pf09G244090 | Isoprenol/Carverol dehydrogenase | |
| | pf04G094750 | Aflatoxin dehydrogenase-like subtype | |
| | pf09G244170 | Carverol dehydrogenase | |
| | pf10G253440 | Carotene $\varepsilon$-monooxygenase | |
| *P. philadelphica* Tieba | pf07G186580 | Chalcone flavonoid isomerase 3 subtype | |
| | pf05G125540 | Flavanone 3-hydroxylase | |
| | pf08G220770 | Flavonol synthase/Flavanone 3-hydroxylase | Flavonoid |
| | pf08G220750 | Flavonol synthase/Flavanone 3-hydroxylase | |
| | pf09G224140 | Anthocyanin synthase | |
| | pf01G020090 | Anthocyanin 3-O-glucosyltransferase | Flavonol |
| | pf11G295430 | Anthocyanin 3-O-glucosyltransferase 7 | |
| | pf09G237080 | Abscisic acid 8$'$-hydroxylase | |
| | pf06G177100 | Zeaxanthin epoxide enzyme | |
| | pf06G169640 | (9-cis-epoxycarotenoid dioxygenase | |
| | pf02G043370 | $\beta$-carotene hydroxylase | |
| | pf09G235870 | Abscisic acid 8$'$-hydroxylase | |
| | pf12G298210 | Monooxygenase | |
| | pf12G301650 | Decaisotriol dehydrogenase-like | |
| | pf04G091760 | Abscisic acid 8$'$-hydroxylase | Carotenoid |
| | pf11G280920 | Short-chain dehydrogenase reductase | |
| | pf03G054950 | Decaisotriol dehydrogenase-like | |
| | pf08G205980 | Shedding aldehyde oxidase-like subtype | |
| | pf10G253440 | Carotene $\varepsilon$ monooxygenase | |
| | pf08G205960 | Indole-3-acetaldehyde oxidase | |
| | pf11G283010 | $\beta$-carotene hydroxylase | |
| | pf12G300180 | Decaisotriol dehydrogenase-like | |
| | pf06G161300 | Short-chain dehydrogenase reductase 4 | |

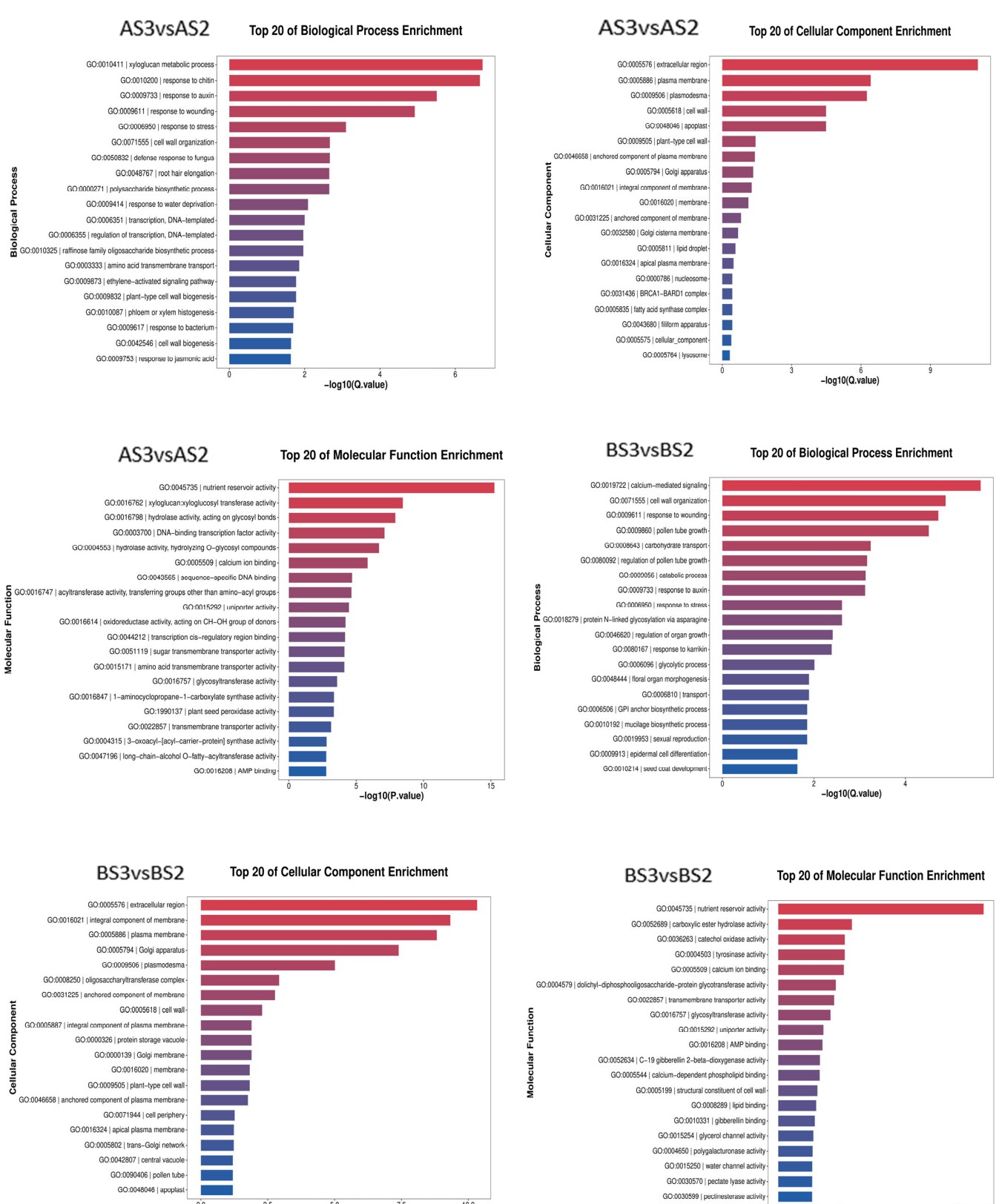

**Figure 6.** *Cont.*

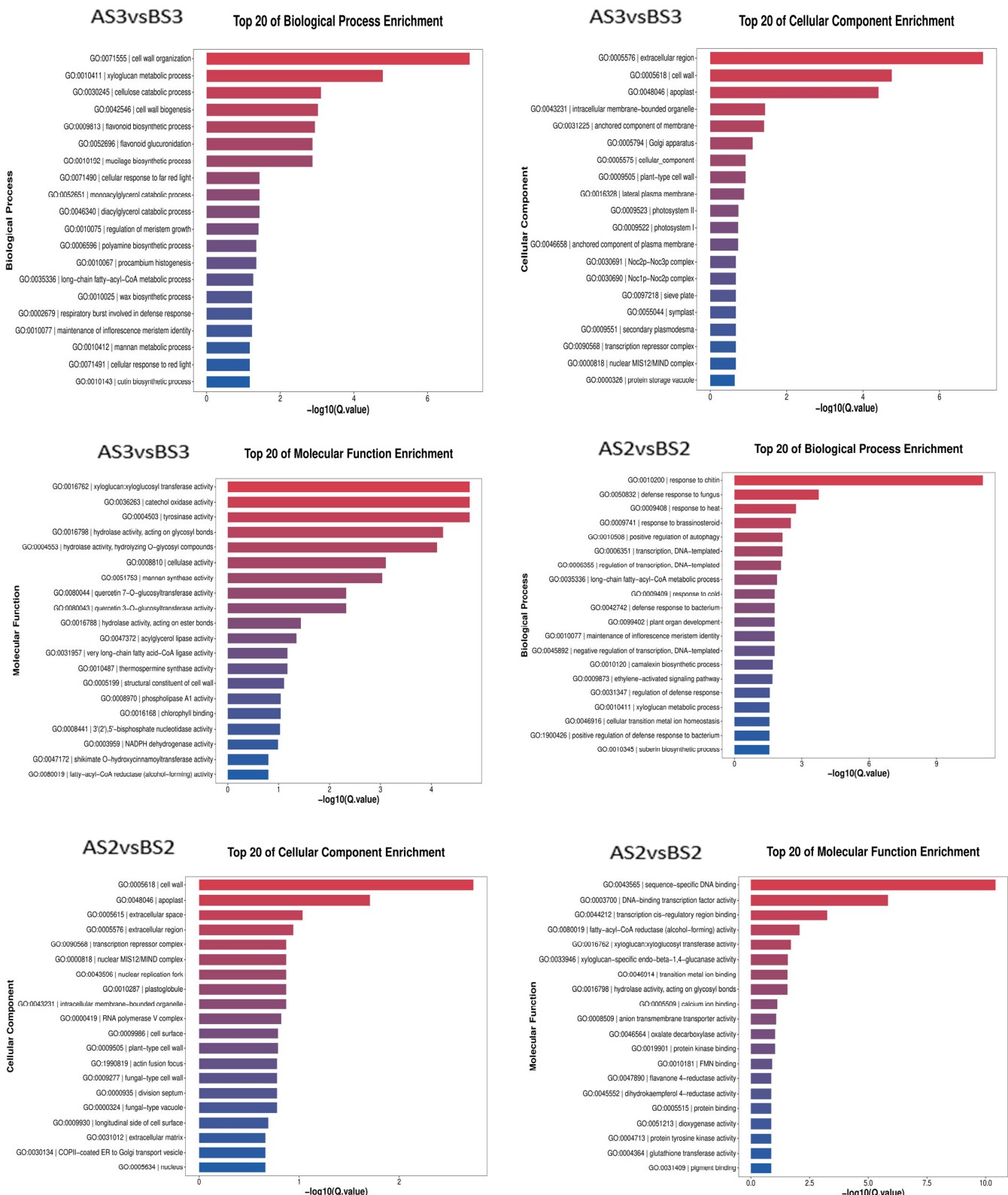

**Figure 6.** Bar chart showing GO enrichment of the differentially expressed genes.

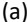

(a)

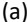

**Figure 7.** *Cont.*

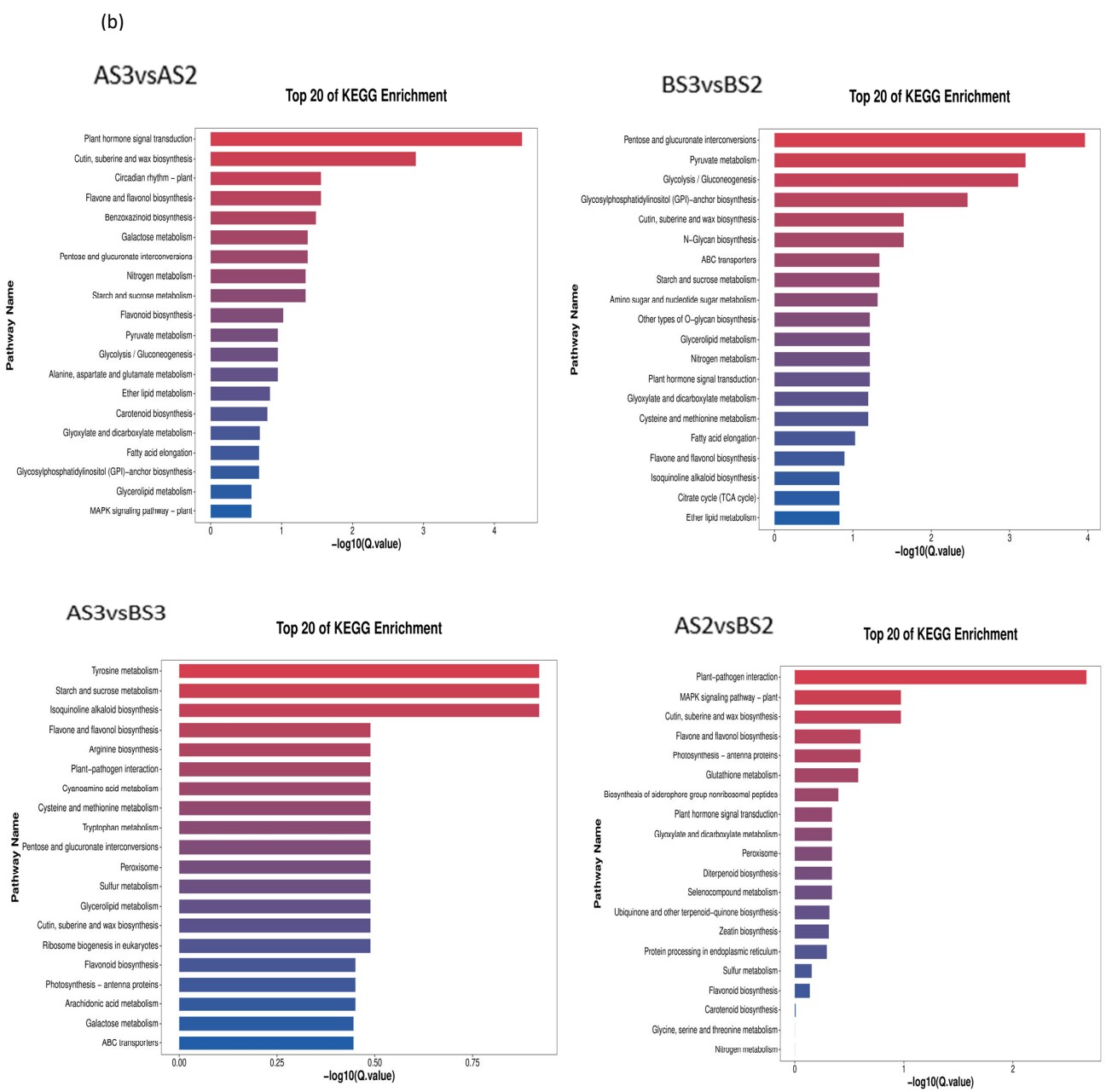

**Figure 7.** Circle diagram (**a**) and bar chart (**b**) showing KEGG enrichment of the differentially expressed genes.

### 3.2.4. Mining of Genes Related to Flavonoids, Flavone and Flavonol, and Carotenoid Biosyntheses

Further in-depth analysis of flavonoids, flavone and flavonol, and carotenoid biosynthesis pathways identified a total of 343 (AS2 vs. BS2), 553 (AS3 vs. BS3), 2275 (AS3 vs. AS2) and 2815 (BS3 vs. BS2) TFs; these included 1177 up-regulated and 1638 down-regulated TFs for BS3 vs. BS2; 1332 up-regulated and 943 down-regulated TFs for AS3 vs. AS2; 127 up-regulated and 426 down-regulated TFs for AS3 vs. BS3; 100 up-regulated and 243 down-regulated TFs for AS2 vs. BS2. Notably, most of these identified TFs belonged to the AP2/ERF, bHLH, MYB, WRKY, GATA, NAC, or bZIP family.

As shown in Tables 7–9, two key TFs (GeBP and STAT) were found involved in anthocyanin synthesis inhibition, which might regulate the inhibitory effects of pf05G124640 (dihydroflavonol 4-reductase) and pf09G224140 (anthocyanin synthase); one key TF (HSF) regulated pf01G020090 (anthocyanin 3-O-glucosyltransferase) in the flavonoid and flavonol

synthesis pathway; two key TFs (NAC and G2-Like) regulated pf10G255070 (cleaved isocastanol dehydrogenase) and pf09G237080 (abscisic acid 8′-hydroxylase) and played important roles in carotenoid biosynthesis.

**Table 7.** Differentially expressed genes related to flavonoid biosynthesis.

| Comparison Groups | Gene Name | Description | Regulation | Transcription Factors |
|---|---|---|---|---|
| AS3 vs. AS2 | pf05G125540 | Hesperetin 3-hydroxylase | Down | Dof |
| | pf07G186580 | Chalcone-flavonoid isomerase | Down | NAC |
| | pf08G220750 | Flavonol synthase/flavanone 3-hydroxylase | Down | STAT |
| | pf08G220770 | Flavonol synthase/flavanone 3-hydroxylase | Down | STAT |
| | pf09G224140 | Anthocyanin synthase | Down | STAT |
| BS3 vs. BS2 | pf05G124640 | Dihydroflavonol 4-reductase | Down | GeBP |
| | pf05G125540 | Flavanone 3-hydroxylase | Down | Dof |
| | pf07G186580 | Chalcone-flavonoid isomerase | Down | NAC |
| | pf08G220750 | Flavonol synthase/flavanone 3-hydroxylase | Down | STAT |
| | pf08G220770 | Flavonol synthase/flavanone 3-hydroxylase | Down | STAT |
| | pf09G224140 | Anthocyanin synthase | Down | STAT |
| | pf09G234590 | Biliverdin dioxygenase | Down | STAT |
| AS3 vs. BS3 | pf05G124640 | Dihydroflavonol 4-reductase | Down | GeBP |
| | pf09G224140 | Anthocyanin synthase | Down | STAT |
| AS2 vs. BS2 | pf05G124640 | Dihydroflavonol 4-reductase | Down | GeBP |
| | pf09G224140 | Anthocyanin synthase | Down | STAT |

**Table 8.** Differentially expressed genes related to flavonoid and flavonol biosynthesis.

| Comparison Groups | Gene Name | Description | Regulation | Transcription Factors |
|---|---|---|---|---|
| AS3 vs. AS2 | pf01G020090 | Anthocyanin 3-O-glucosyltransferase | Down | HSF |
| | pf11G295430 | Anthocyanin 3-O-glucosyltransferase | Down | GRAS |
| BS3 vs. BS2 | pf01G020090 | Anthocyanin 3-O-glucosyltransferase | Down | HSF |
| AS3 vs. BS3 | pf01G020090 | Anthocyanin 3-O-glucosyltransferase | Down | HSF |
| AS2 vs. BS2 | pf01G020090 | Anthocyanin 3-O-glucosyltransferase | Down | HSF |

**Table 9.** Differentially expressed genes related to carotenoid biosynthesis.

| Comparison Groups | Gene Name | Description | Regulation | Transcription Factors |
|---|---|---|---|---|
| AS3 vs. AS2 | pf04G091760 | Abscisic acid 8′-hydroxylase | Up | G2-like |
| | pf06G161300 | Short-chain dehydrogenase reductase | Down | NAC |
| | pf08G205960 | Indole-3-acetaldehyde oxidase | Down | LBD |
| | pf09G235870 | Abscisic acid 8′-hydroxylase | Up | G2-like |
| | pf09G237080 | Abscisic acid 8′-hydroxylase | Up | G2-like |
| | pf10G253440 | Carotene ε-monooxygenase | Down | bHLH |
| | pf11G280920 | Short-chain dehydrogenase reductase | Up | NAC |
| BS3 vs. BS2 | pf04G094750 | Aflatoxin dehydrogenase | Down | NAC |
| | pf09G235870 | Abscisic acid 8′-hydroxylase | Up | G2-like |
| | pf09G237080 | Abscisic acid 8′-hydroxylase | Up | G2-like |
| | pf09G244090 | Isopentenol dehydrogenase | Down | bHLH |
| | pf09G244170 | Isopentenol dehydrogenase | Down | NAC |
| | pf10G253440 | Carotene ε-monooxygenase | Down | bHLH |
| | pf10G255070 | Divinyl ether synthase | Down | NAC |
| AS3 vs. BS3 | pf10G255070 | Divinyl ether synthase | Up | NAC |
| AS2 vs. BS2 | pf09G237080 | Abscisic acid 8′-hydroxylase | Down | G2-like |

### 3.2.5. qRT-PCR Validation of NGS Results

To validate the accuracy and reliability of the RNA-Seq data, 12 DEGs (pf05G124640, pf09G224140, pf05G137060, pf01G020090, pf05G120860, pf06G177100, pf06G169640, pf09G237080, pf05G143230, pf01G016330, pf11G267170 and pf06G179690) were randomly selected for qRT-PCR expression analyses. As shown in Figures 8 and 9, the trend of qRT-PCR is consistent with NGS (Tables 10 and 11). The differential expression analyses based on RNA-Seq are reliable and accurate.

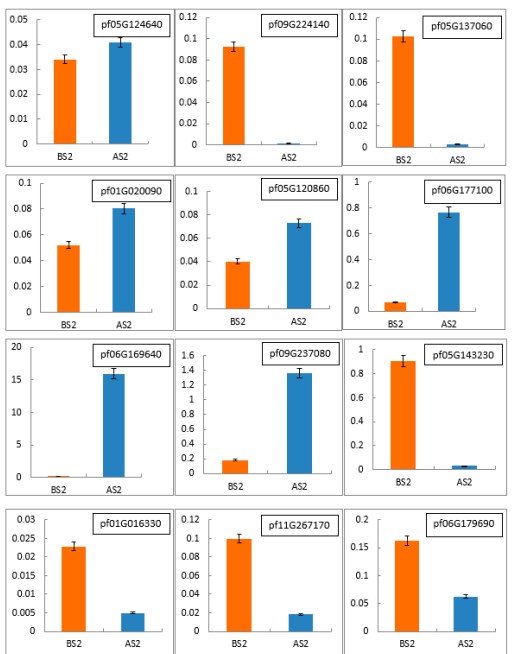
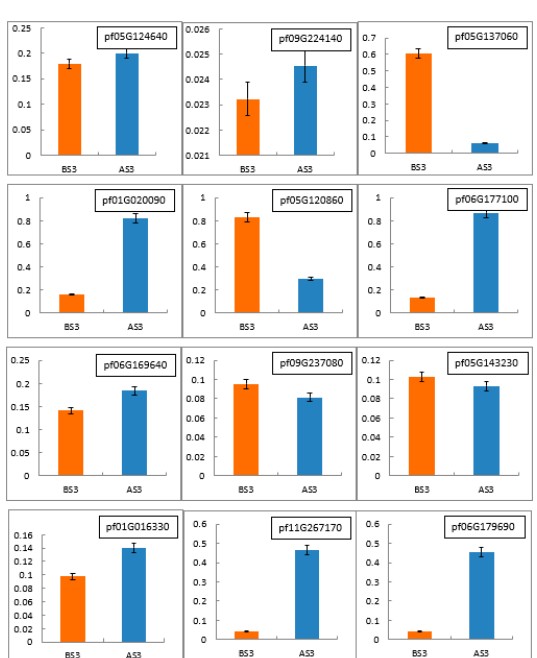

**Figure 8.** Real-time quantitative PCR (RT-qPCR) was conducted to verify the expression levels of differentially expressed genes in Tieba (A) and Qingjin (B) during the same flowering stages, including the initial flowering stage (S2) and full flowering stage (S3).

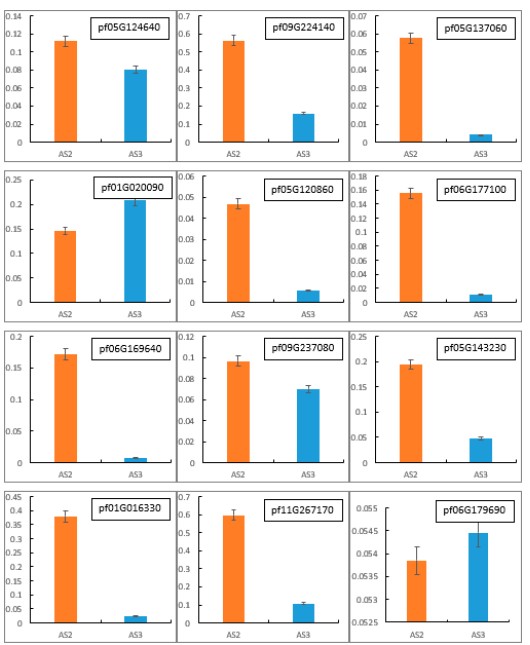
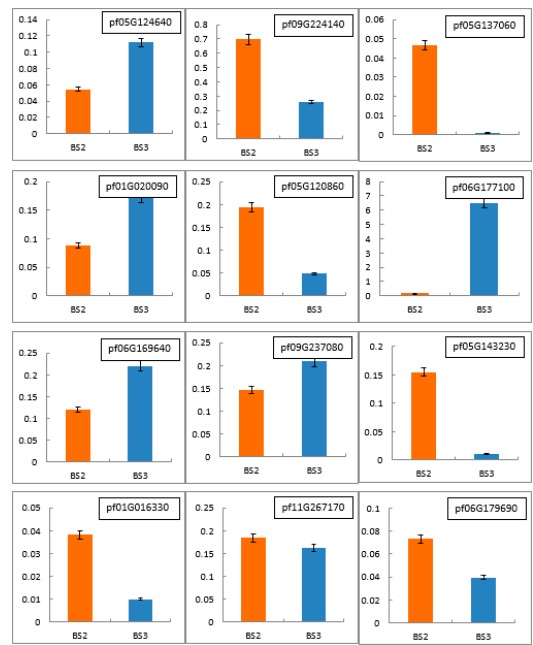

**Figure 9.** Real-time quantitative PCR (RT-qPCR) was conducted to verify the expression levels of differentially expressed genes in Tieba (A) and Qingjin (B) during different flowering stages, including the initial flowering stage (S2) and the full flowering stage (S3).

**Table 10.** The expression levels of differentially expressed genes in Tieba A.

| Gene_Name | FPKM. A_S2_1 | FPKM. A_S2_2 | FPKM. A_S2_3 | FPKM. A_S3_1 | FPKM. A_S3_2 | FPKM. A_S3_3 |
|---|---|---|---|---|---|---|
| pf01G016330 | 137.83 | 95.08 | 137.20 | 15.39 | 12.22 | 22.80 |
| pf01G020090 | 21.22 | 15.23 | 15.81 | 0.52 | 0.24 | 0.10 |
| pf05G120860 | 5.24 | 6.28 | 4.51 | 25.08 | 17.11 | 25.89 |
| pf05G124640 | 0.12 | 0.16 | 0.21 | 0.34 | 0.10 | 0.26 |
| pf05G137060 | 17.29 | 7.12 | 12.47 | 0.32 | 0.27 | 0.06 |
| pf05G143230 | 97.28 | 58.98 | 89.15 | 2.93 | 1.03 | 1.52 |
| pf06G169640 | 15.47 | 7.32 | 19.59 | 56.88 | 72.64 | 69.55 |
| pf06G177100 | 4.90 | 4.75 | 4.09 | 48.37 | 46.04 | 41.24 |
| pf06G179690 | 24.93 | 20.92 | 21.57 | 6.69 | 7.97 | 8.74 |
| pf09G224140 | 4.25 | 3.18 | 4.21 | 1.58 | 2.03 | 1.02 |
| pf09G237080 | 0.57 | 1.03 | 0.79 | 42.06 | 33.00 | 33.05 |
| pf11G267170 | 57.51 | 58.49 | 41.90 | 9.35 | 18.56 | 31.15 |

**Table 11.** The expression levels of differentially expressed genes in Qingjin B.

| Gene_Name | FPKM. B_S2_1 | FPKM. B_S2_2 | FPKM. B_S2_3 | FPKM. B_S3_1 | FPKM. B_S3_2 | FPKM. B_S3_3 |
|---|---|---|---|---|---|---|
| pf01G016330 | 289.09 | 264.30 | 220.95 | 23.24 | 22.72 | 30.06 |
| pf01G020090 | 57.01 | 58.46 | 38.68 | 3.03 | 1.39 | 5.39 |
| pf05G120860 | 15.91 | 15.51 | 28.10 | 17.91 | 17.05 | 35.35 |
| pf05G124640 | 293.83 | 295.91 | 229.43 | 26.21 | 15.19 | 36.54 |
| pf05G137060 | 175.92 | 190.60 | 107.30 | 15.54 | 4.24 | 16.09 |
| pf05G143230 | 105.13 | 86.95 | 72.13 | 2.95 | 2.59 | 4.28 |
| pf06G169640 | 53.47 | 36.49 | 56.95 | 46.96 | 43.04 | 76.05 |
| pf06G177100 | 10.70 | 21.88 | 29.88 | 23.83 | 34.22 | 32.77 |
| pf06G179690 | 25.27 | 28.91 | 22.19 | 3.62 | 4.62 | 6.03 |
| pf09G224140 | 100.57 | 102.84 | 87.72 | 9.12 | 7.99 | 9.99 |
| pf09G237080 | 2.54 | 2.13 | 11.36 | 14.99 | 10.24 | 27.84 |
| pf11G267170 | 66.09 | 54.08 | 47.89 | 6.90 | 9.85 | 6.41 |

## 4. Discussion

*P. philadelphica* stands out as one of the distinctive fruits in Northeast China, appreciated for its medicinal, edible, and ornamental values [1,2]. The majority of corollas exhibit a yellowish or yellow hue, and the flowering and fruiting phases typically span from May to November [27]. Native to the Americas, *P. philadelphica* is now cultivated in the northeastern provinces of China, including Jilin and Heilongjiang [2]. Variations in petal coloration typically arise from differences in the types and concentrations of pigments present. Previous studies have shown that high contents of anthocyanins in petals result in darker-colored flowers [28,29]. Six pigments are commonly found in colored plants, including anthocyanins, delphinidin, paeoniflorin, maleic glycoside, pelargonidin and petunidin [30], among which anthocyanins contribute to the reddish-purple color, delphinidin contributes to the bluish-red or purple color, and pelargonidin contributes to the orange and red color [31]. Hsueh et al. demonstrated that anthocyanins, pelargonidin, and delphinidin are the main components of strawberry safflower [32].

For this study, we selected *P. philadelphica* var. Tieba (yellow flower) and *P. philadelphica* var. Qingjin (yellow-purple flower) as representative subjects to investigate the differences in flower coloration. We focused on two stages of flower development, the initial and full flowering stages. Through HPLC, we analyzed the associated pigment components and determined their relative concentrations. We found that the contents of four anthocyanin glycosides (i.e., malvidin, pelargonidin, delphinidin and cyanidin) detected in Tieba (A) were less than those of the standard samples, suggesting that they were not related to petal coloration of *P. philadelphica* In contrast, the presence of delphinidin, cyanidin and malvidin pigments and the absence of pelargonidin in Qingjin (B) indicated that the

former three anthocyanin glycosides primarily contributed to the the variation in petal coloration in Qingjin. This further suggested that delphinidin, cyanidin, and malvidin are the predominant anthocyanin components responsible for the purple coloration observed in the petals of *P. philadelphica*.

### 4.1. Influence of Key Enzyme Genes on Petal Coloration of P. philadelphica

We identified the key genes associated with petal coloration in two varieties of *P. philadelphica*. We further focused on the three biosynthetic pathways deemed most relevant for analysis. The findings revealed that the differences in petal coloration between the two *P. philadelphica* varieties stemmed from the negative regulation of the key genes within the flavonoid biosynthesis pathway, specifically pf05G124640 (dihydroflavonol 4-reductase) and pf09G224140 (anthocyanin synthase). Annotation results of these two genes corroborate previous reports of gene regulation within anthocyanin synthesis pathways in other varieties [33].

The enzyme known to influence fruit color by catalyzing the formation of the initial stable intermediate anthocyanin in strawberries is anthocyanin-3-O-glucosyltransferase, FaGT1. It converts uridine diphosphate (UDP)-glucose to anthocyanin and, to a lesser extent, facilitates flavonol glycosylation to produce the corresponding 3-O-glucoside [34]. Consistently in our study, during the comparative gene analysis of the two *P. philadelphica* varieties, significant differences were observed in the genes encoding key enzymes within the flavonoid and flavonol biosynthesis pathways. Specifically, variations were noted in the UDP-glucose converting enzyme and the 3-O-glucosidase, indicating a positive correlation between these genes and anthocyanin formation.

### 4.2. TFs Associated with Petal Coloration in P. philadelphica

Pigment biosynthesis involves a complex network, where the function is not solely attributed to a single structural gene or TF, but rather relies on the collective action of multiple TFs and structural genes. Apart from structural genes, TFs such as MYB, bHLH, WD proteins, and MADS-box proteins play important roles in anthocyanin biosynthesis [35]. Previous studies have indicated that PpNAC1 promotes anthocyanin accumulation by activating the transcription of PpMYB10.1, whereas the expression of PpNAC2 is repressed by PpSPL1 [36]. This was likewise predicted by our transcriptome analysis that revealed a multitude of TFs, among which 12 TFs were identified based on their expression levels and correlation annotations with three pigment-associated biosynthesis pathways. These TFs are likely to play crucial roles in the phenotypic expression of petal coloration.

Among these TFs, NAC exhibited two different expression patterns within the flavonoid biosynthesis pathway (map00941) and the carotenoid biosynthesis pathway (map00906). NAC regulation of pf07G186580 trending in the opposite direction of total anthocyanin content, and NAC regulation of pf04G094750 trending in the opposite direction of carotenoid content positively. The NAC (NAM, ATAF1, ATAF2, and CUC2) family is a unique group of TFs found in higher plants, comprising one of the largest TF families in the plant genome with over 100 characterized members. These TFs play vital roles in various physiological processes, including the regulation of plant secondary growth and hormone signaling pathways.

### 5. Conclusions

The pigment contents in both *P. philadelphica* varieties increased gradually with the reproductive period, with Qingjin petals being rich in anthocyanins, specifically malvidin, pelargonidins, delphinidin, and cyanidin, while carotenoids predominated in Tieba petals, with β-carotene contributing significantly to color intensity. Key enzyme genes including pf05G124640 (dihydroflavonol 4-reductase), pf09G224140 (anthocyanin synthase), pf01G020090 (anthocyanin 3-O-glucosyltransferase), pf10G255070 (cleavage isocastanol dehydrogenase) and pf09G237080 (abscisic acid 8′-hydroxylase) were identified as crucial determinants of petal coloration differences between the two varieties. They significantly influenced flavonoid (map00941), flavonoid and flavonol (map00944), and

carotenoid (map00906) biosynthetic pathways. Additionally, a comparison of developmental stages revealed the involvement of key TFs such as GeBP, STAT, HSF, NAC, and G2-like in regulating various enzymes and pathways associated with anthocyanin and carotenoid biosynthesis, thereby elucidating complex molecular mechanisms underlying petal coloration in *P. philadelphica*.

**Author Contributions:** Data curation: J.Z. (Jingying Zhang) and W.Z., Formal analysis: J.Z. (Jingying Zhang) and W.Z., Investigation: S.T., D.W., and N.L., Methodology: H.W., C.W., and X.Z., Resources: S.T., J.Z. (Jiaming Zhu), and S.M., Supervision: H.Q. and H.H., Writing—original draft: J.Z. (Jingying Zhang) and W.Z., Writing—review and editing: H.Q. and H.H. All authors have read and agreed to the published version of the manuscript.

**Funding:** This research received no external funding.

**Data Availability Statement:** All RNA-Seq raw sequence data files are available in the NCBI SRA Database (https://dataview.ncbi.nlm.nih.gov, (accessed on 24 February 2024)) under accession numbers SRR28082792, SRR28082794, SRR28082799, SRR28082791, SRR28082793, SRR28082797, SRR28082795, SRR28082798, SRR28082801 and SRR28082793.

**Conflicts of Interest:** The authors declare no conflicts of interest.

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
