# Peer review of "Exploring the Mechanisms Underlying Petal Pigmentation Differences in Two Cultivars of Physalis philadelphica Based on HPLC and NGS"

_horticulturae, doi:10.3390/horticulturae10050507_

Round 1

Reviewer 1 Report

Comments and Suggestions for Authors

In this manuscript Zhang and collaborators used HPLC and further RNAseq to compare pigments between different accessions of Physalis philadelphica. Albeit the main scope for tomatillo cultivation does not reside in its haestetical characteristics, flowers color represents an interesting and actracting characteristic.

Although carotenoid and anthocyanin pathways have been described in several species and much of their regulative pathways are known, such work is interesting as it aims at identifying the differencies in the regulation of both pathways in tomatillo.

That said, I would recommend an intense and thoroughful revision of the paper, as in some parts it is difficult to follow or, even worse, the authors sometimes indicate two opposite metabolites/pathways/etc as the correct one. For instance, in the abstract you state (lines 28-31) that both anthocyanins and carotenoids are "the main substance of petal coloring", which of course cannot be: please amend.

Below you can find some comments to be addressed.

In the introduction I ask to add references as many of your statements are unreferenced

- line 47: "yellow mushrooms" or "foreign mushrooms" are perhaps typical of chinese regions, but in the rest of the world the plant is also known as "tomatillo"; please add. I would also describe the origin of the plant; regarding its uses, please provide some references.

- phytochromes are light receptors, not pigments

- refs 1-2 do not refer to Physalis philadelphica

- please add pictures of the flowers to have an idea what you are discussing about

- what are 12 samples referred to? I thought there were 2 samples (varieties) analuzed at 2 different time points. Replicates?

- section 2.1 is difficult to understand: how many samples, how many replicates?

- section 2.2.4: which reference genome? whole paragraph very difficult to understand. How did you estimate DEGs? which software did you use?

- how did you identify genes involed in anthocyanin/carotenoid biosynthesis and regulation? Was there an annotation or did you annotate them independently? If this was the case, how?

- statistics used to deem for different accumulation of metabolites?

- section 3.1.3: name of metabolites is different with respect to Table 4

- section 3.2: not clear. Supplementary Table is necessary to report sequencing metrics. Text not clear, has to be deeply revised

-lines 249-273: more similar to Figure captions and Material and methods than to Results section

- 9 references (19-27) should be distributed as they are wrongly placed. Moreover, you should enrich the discussion section by adding correct references in the right place.

- it is known that members of biosynthetic pathways are modulated (by TF, feedback, etc): did you find correspondences/confirmations in your data?

- Figure 6 is central for your discussion but I could not revise that part as I could not see the text

- The discussion part where you discuss about TF expression needs to be restructured. You discuss much about NAC interactions, but this family is not much involved in color synthesis or, better, there are many other TFs involved that you did not discuss. Even if you could not spot differences, you'd better report that. Which NAC did you identify as differentially expressed? Just one or more than one?

- Figures 5-6: quality of the pictures is not high enough to appreciate what is written in the text

- Figures 7-8: in the caption is missing what the bars refer to; moreover it would be more informative whether you could add the FPKM values of the same genes as derived from RNAseq experiments

- Tables 6-8: I do not understand what the genes indicate refer to: are they enzymes (as in the "description" column) or TFs (as in "Transcription Factors" column)?

Comments on the Quality of English Language

The abstract is very difficult to follow (especially lines 24-28), maybe for incorrect cut-copy-paste or other issues; in general, the whole manuscript has to be carefully and thoroughfully revised.

Author Response

Hello, expert! Thank you for your valuable revision suggestions. I have revised each suggestion accordingly, and the updated files have been uploaded.

Reviewer 2 Report

Comments and Suggestions for Authors

In this manuscript, the authors investigated the mechanisms involved in petal pigmentation differences in two cultivars of Physalis philadelphica using different experimental approaches. The results appear promising; however, the presentation of the paper is deficient requiring extensive improvements. Some points are detailed below.

The abstract needs to be rewritten to be more clear and direct regarding objective and the data presentation. It also needs a general conclusion.

Lines 47, 50 and 84; see the correct use of capital or lower case letters after “L.”

Line 58. The word “color” is repetitive.

Line 59 and 60. The correct name of the “order” is Caryophyllales and not Caryophyllale.

The final paragraph of introduction needs to be revised clarifying the aim of manuscript. Use a language more direct.

The experimental design is not clear, please revise.

Line 122. What means this point: “file. Format”

Line 130. Separate “differentiallyexpressed”

The authors state that “primers were synthesized by Primer 6.0 software…” however it is not correct. This program is used to design primers, not to perform synthesis.

In RT-qPCR analyses, the authors inform that they used the gene Pp.GAPDH as internal control. Question: Have the authors evaluated the Pp.GAPDH expression stability in the samples to ensure reliable quantitative analyses?

In result section, capital letters are incorrectly used in several situations…

Line 178 and 179. The authors state that carotenoids in “… Tieba initial flowering stage (AS2) > Tieba full flowering stage (AS3)” but these data no have statistic differences!!! Also, in others situations in this same section authors sustain for differences when statistics support similarities…

In results, the topic “3.1.3. Analysis of anthocyanin content” is more or less similar to “3.1.1. Analysis of total anthocyanin content”. This cause confusion! Thus, I suggest changing the topic 3.1.3 including more specific details.

 I do not understand the results from the table 5. The data 108.046 and 45.609 are statistically similar???

In line 121, the authors state that “..A total of 56.30 Gb of Clean Data was obtained from the cDNA library construction”…  while in the line 223 they write “and the Clean Data of all the samples reached 5.95 Gb…” Thus, what is correct?

Figures 3, 5, 6 are not visible needing improvements.

With regard to data availability, no information was given about how the raw data will be available to the public.

The English needs extensive revision.

Comments on the Quality of English Language

The English needs extensive revision.

Author Response

(The authors gave the same response as above.)

Reviewer 3 Report

Comments and Suggestions for Authors

The article is interesting because it shows the relationship between DNA and color formation in Physalis philadelphica.
What the authors seek with this study is to identify plants' metabolic information and DNA involvement.
In the introduction, it is recommended that the authors put information on the development and formation of color and pigments in plants, in particular Physalis philadelphica, to have an overview of how it develops and its dependence on metabolism and the relationship with DNA.
The results presented in some figures are illegible as in Figure 3,
figure 5, figure 6.
It is suggested to the authors to put more legible figures in the manuscript because some figures are distorted when magnified to observe them.

Comments on the Quality of English Language

The article is interesting because it shows the relationship between DNA and color formation in Physalis philadelphica.
What the authors seek with this study is to identify plants' metabolic information and DNA involvement.
In the introduction, it is recommended that the authors put information on the development and formation of color and pigments in plants, in particular Physalis philadelphica, to have an overview of how it develops and its dependence on metabolism and the relationship with DNA.
The results presented in some figures are illegible as in Figure 3,
figure 5, figure 6.
It is suggested to the authors to put more legible figures in the manuscript because some figures are distorted when magnified to observe them.

Author Response

Hello, expert! Thank you for your valuable revision suggestions. The figures in the manuscript were not very clear, so I have re-uploaded them.

Please see the detailed responses in the attached file

Round 2

Reviewer 1 Report

Comments and Suggestions for Authors

I thank the authors for having addressed my comments; in my view the manuscript has now improved; however a further review is necessary as some aspects have not been solved.

- Lines 28-29: these represent the take-home message, they should be moved to the end of the abstract

- Lines 80-86: I appreciate the modifications you introduced regarding the composition of the samples used for analysis: however, I feel it would be useful, for clarity's sake, to write that the analyses were performed on 3 biological replicates.

- Please add a reference or a link to identify the genome assembly used for transcript quantifications 

- You still have to clarify whether you used statistics (which software?) to identify differentially expressed genes and metabolites and to add the relevant text in the Methods section

- did you use a software to identify Differentially Expressed Genes? If this is the case, please add it in the Methods section

- you should add the software you used to produce the Figures

- Lines 216-219: if you are not going to show the results, please add "data not shown" in the text. Did you use your novel predictions for Differential Gene Analysis or you used the official annotation (still have to add it in the text)

- Line 225: you wrote AS2 vs BS2 twice

- Lines 231-249 are covered by Figure 4: please reformat this part.

- Lines 308-310: it is not clear what you did: did you perhaps perform enrichment analysis using the DEGs you identified in the previous section using GO terms and KEGG entries?

- Supplementary materials have not been referenced in the main text: please do it where needed (for example, Supplementary Tables 2 and 3 (you wrote "2" twice) should be cited in section 3.2.5

- lines 479, 492, etc: those are varieties or cultivars, not species

- line 497: those are pigments, not phytochromes

- Ref 34 is not accessible and can't be checked as it is inserted. I think there are other papers that can be cited

- Figure 2: missing

- Figure 4: pls move it appropriately

- Figure 5: how did you perform the clustering? This is not written in the Methods section

- Figures 7 and 9 are cut

- Tables 2-3-4-5: I assume the values represent means of the measurements of the replicates; please add it in the caption and add the relative standard deviations. In Table 5, I would add the same number of decimals

Author Response

Hello, Expert. Thank you very much for your valuable suggestions for revision. I have made revisions to each suggestion, and the content is as follows:

-Lines 28-29: these represent the take-home message, they should be moved to the end of the abstract
Thank you very much for your valuable suggestions for revision. The content related to lines 28-29 has been moved to the end of the abstract, as specified in lines 39-41, which are highlighted in red in the text.

- Lines 80-86: I appreciate the modifications you introduced regarding the composition of the samples used for analysis: however, I feel it would be useful, for clarity's sake, to write that the analyses were performed on 3 biological replicates.
Thank you very much for your valuable suggestions for revision. The two-stage selection (S2: Initial flowering stage; S3: Full flowering stage) is shown in Figure 1, with three biological replicates as specified in lines 85-86, highlighted in red in the text.

- Please add a reference or a link to identify the genome assembly used for transcript quantifications
Thank you very much for your valuable suggestions for revision. The relevant link has been added in the text, specifically on line 128-129, and it has been highlighted in red in the text.

- You still have to clarify whether you used statistics (which software?) to identify
differentially expressed genes and metabolites and to add the relevant text in the Methods section
Thank you very much for your valuable suggestions for revision. The analysis of
metabolite data was performed using DPS7.5 software, which has been added to the Methods section. Specific details can be found in line 114, highlighted in red in the text.

-did you use a software to identify Differentially Expressed Genes? If this is the case, please add it in the Methods section
The identification of differentially expressed genes was conducted using the DESeq2 software, as elaborated in the Methods section at line 134, which has been highlighted in red in the text.

- you should add the software you used to produce the Figures
Thank you very much for your valuable suggestions for revision. Venn diagrams, volcano plots and heatmaps were generated using the Lianchuan Bio Cloud Platform. Specific details can be found in line 144-145, highlighted in red in the text.

- Lines 216-219: if you are not going to show the results, please add "data not shown" in the text. Did you use your novel predictions for Differential Gene Analysis or you used the official annotation (still have to add it in the text)
Thank you very much for your valuable suggestions for revision. Perform differential gene analysis using official annotations. The relevant content has been added to the text; please refer to line 235-238, which has been marked in red in the text.

- Line 225: you wrote AS2 vs BS2 twice
I'm sorry, I didn't check thoroughly before and repeated AS2 vs BS2. The redundant content "AS2 vs BS2" has been removed from the text, as seen in lines243-244, highlighted in red.

- Lines 231-249 are covered by Figure 4: please reformat this part.
I'm sorry, I'm having trouble understanding how to format this part, possibly due to a computer issue. There seems to be a discrepancy between lines 231-249 that I found. The part "The highest number of DEGs were found from S3 vs S2 comparisons" refers to the content of Figure 3. This part "In addition, overlapping DEGs were identified among AS3 vs AS2, BS3 vs BS2, AS3 vs BS2, and AS2 vs BS2 comparison groups, indicating their potential key roles in petal color expression across different varieties" refers to the content of Figure 2.

- Lines 308-310: it is not clear what you did: did you perhaps perform enrichment analysis using the DEGs you identified in the previous section using GO terms and KEGG entries?
YES, expert. GO enrichment and KEGG analysis of petal coloring-related DEGs in Physalis philadelphica L. were conducted based on the DEGs identified in the previous section.

- Supplementary materials have not been referenced in the main text: please do it where needed (for example, Supplementary Tables 2 and 3 (you wrote "2" twice) should be cited in section 3.2.5
I apologize for the repetition of the table numbers, it has been revised. The relevant tables are referenced in Section 3.2.5, with specific content found on pages 465-485, highlighted in red in the text.

- lines 479, 492, etc: those are varieties or cultivars, not species
I apologize, the relevant content has been corrected. The term "cultivars" has been changed in the specific lines. The specific content can be found in lines 516, 519, 523, 529, 555 and 563, highlighted in red in the text.

- line 497: those are pigments, not phytochromes
The relevant content has been changed to "Pigment," as specified in line 534, which is highlighted in red in the text.

- Ref 34 is not accessible and can't be checked as it is inserted. I think there are other papers that can be cited
Thank you for your valuable suggestions for modification. Reference 34 has been modified, and the specific content can be found in lines 647-648, marked in red in the text.

- Figure 2: missing
The specific content of Figure 2 can be found in lines 253-268.

- Figure 4: please move it appropriately
Thank you for your valuable suggestions for modification. Figure 4 has been appropriately moved to the right, as described in lines 277-299.

- Figure 5: how did you perform the clustering? This is not written in the Methods section
Thank you for your valuable suggestions for modification. The clustering-related content has been supplemented in the Methods section, with specific details provided in lines 134-142, highlighted in red in the text.

- Figures 7 and 9 are cut
I'm sorry, Figures 7 and 9 have been adjusted, and the specific content can be found in lines 368-393 and 451-464.

- Tables 2-3-4-5: I assume the values represent means of the measurements of the replicates; please add it in the caption and add the relative standard deviations. In Table 5, I would add the same number of decimals
Thank you for your valuable suggestions for modification. The titles of Tables 2-3-4-5 have been amended to include the information "These values represent the average of repeated measurements." , the relative standard deviation has been added to the table, the same number of decimal places has been added in Table 5. Specific details can be found in lines 178-180, 192-194, 212-214 and 224-226 highlighted in red in the text.

The specific content has also been uploaded in the attachment for your review.

Reviewer 2 Report

Comments and Suggestions for Authors

The authors accepted the large majority of my suggestions and the manuscript has been significantly improved. Thus, I agree with its acceptance for publication.

Author Response

Hello, Expert. Thank you very much for your approval and for your valuable suggestions for revision recently.